# POLAR TRANSFORMER NETWORKS

**Carlos Esteves, Christine Allen-Blanchette, Xiaowei Zhou, Kostas Daniilidis**
GRASP Laboratory, University of Pennsylvania
{machc, allec, xiaowz, kostas}@seas.upenn.edu

## ABSTRACT

Convolutional neural networks (CNNs) are inherently equivariant to translation. Efforts to embed other forms of equivariance have concentrated solely on rotation. We expand the notion of equivariance in CNNs through the Polar Transformer Network (PTN). PTN combines ideas from the Spatial Transformer Network (STN) and canonical coordinate representations. The result is a network invariant to translation and equivariant to both rotation and scale. PTN is trained end-to-end and composed of three distinct stages: a polar origin predictor, the newly introduced polar transformer module and a classifier. PTN achieves state-of-the-art on rotated MNIST and the newly introduced SIM2MNIST dataset, an MNIST variation obtained by adding clutter and perturbing digits with translation, rotation and scaling. The ideas of PTN are extensible to 3D which we demonstrate through the Cylindrical Transformer Network.

## 1 INTRODUCTION

Whether at the global pattern or local feature level (Granlund, 1978), the quest for (in/equi)variant representations is as old as the field of computer vision and pattern recognition itself. State-of-the-art in "hand-crafted" approaches is typified by SIFT (Lowe, 2004). These detector/descriptors identify the intrinsic scale or rotation of a region (Lindeberg, 1994; Chomat et al., 2000) and produce an equivariant descriptor which is normalized for scale and/or rotation invariance. The burden of these methods is in the computation of the orbit (i.e. a sampling the transformation space) which is necessary to achieve equivariance. This motivated steerable filtering which guarantees transformed filter responses can be interpolated from a finite number of filter responses. Steerability was proved for rotations of Gaussian derivatives (Freeman et al., 1991) and extended to scale and translations in the shiftable pyramid (Simoncelli et al., 1992). Use of the orbit and SVD to create a filter basis was proposed by Perona (1995)and in parallel, Segman et al. (1992) proved for certain classes of transformations there exists *canonical coordinates* where deformation of the input presents as translation of the output. Following this work, Nordberg & Granlund (1996) and Hel-Or & Teo (1996); Teo & Hel-Or (1998) proposed a methodology for computing the bases of equivariant spaces given the Lie generators of a transformation. and most recently, Sifre & Mallat (2013) proposed the scattering transform which offers representations invariant to translation, scaling, and rotations.

The current consensus is representations should be learned not designed. Equivariance to translations by convolution and invariance to local deformations by pooling are now textbook (LeCun et al. (2015), p.335) but approaches to equivariance of more general deformations are still maturing. The main veins are: Spatial Transformer Network (STN) (Jaderberg et al., 2015) which similarly to SIFT learn a canonical pose and produce an invariant representation through warping, work which constrains the structure of convolutional filters (Worrall et al., 2016) and work which uses the filter orbit (Cohen & Welling, 2016b) to enforce an equivariance to a specific transformation group.

In this paper, we propose the Polar Transformer Network (PTN), which combines the ideas of STN and canonical coordinate representations to achieve equivariance to translations, rotations, and dilations. The three stage network learns to identify the object center then transforms the input into log-polar coordinates. In this coordinate system, planar convolutions correspond to group-convolutions in rotation and scale. PTN produces a representation equivariant to rotations and dilations without

---

http://github.com/daniilidis-group//polar-transformer-networks

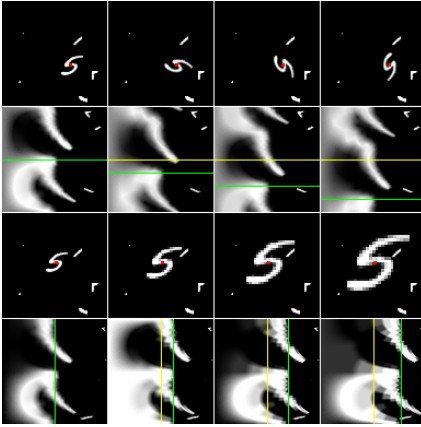

Figure 1: In the log-polar representation, rotations around the origin become vertical shifts, and dilations around the origin become horizontal shifts. The distance between the yellow and green lines is proportional to the rotation angle/scale factor. Top rows: sequence of rotations, and the corresponding polar images. Bottom rows: sequence of dilations, and the corresponding polar images.

the challenging parameter regression of STN. We enlarge the notion of equivariance in CNNs beyond Harmonic Networks (Worrall et al., 2016) and Group Convolutions (Cohen & Welling, 2016b) by capturing both rotations and dilations of arbitrary precision. Similar to STN; however, PTN accommodates only global deformations.

We present state-of-the-art performance on rotated MNIST and SIM2MNIST, which we introduce. To summarize our contributions:

- We develop a CNN architecture capable of learning an image representation invariant to translation and equivariant to rotation and dilation.
- We propose the polar transformer module, which performs a differentiable log-polar transform, amenable to backpropagation training. The transform origin is a latent variable.
- We show how the polar transform origin can be learned effectively as the centroid of a single channel heatmap predicted by a fully convolutional network.

## 2 RELATED WORK

One of the first equivariant feature extraction schemes was proposed by Nordberg & Granlund (1996) who suggested the discrete sampling of 2D-rotations of a complex angle modulated filter. About the same time, the image and optical processing community discovered the Mellin transform as a modification of the Fourier transform (Zwicke & Kiss, 1983; Casasent & Psaltis, 1976). The Fourier-Mellin transform is equivariant to rotation and scale while its modulus is invariant.

During the 80's and 90's invariances of integral transforms were developed through methods based in the Lie generators of the respective transforms starting from one-parameter transforms (Ferraro & Caelli, 1988) and generalizing to Abelian subgroups of the affine group (Segman et al., 1992).

Closely related to the (in/equi)variance work is work in steerability, the interpolation of responses to any group action using the response of a finite filter basis. An exact steerability framework began in Freeman et al. (1991), where rotational steerability for Gaussian derivatives was explicitly computed. It was extended to the shiftable pyramid (Simoncelli et al., 1992), which handle rotation and scale. A method of approximating steerability by learning a lower dimensional representation of the image deformation from the transformation orbit and the SVD was proposed by Perona (1995).

A unification of Lie generator and steerability approaches was introduced by Teo & Hel-Or (1998) who used SVD to reduce the number of basis functions for a given transformation group. Teo and Hel-Or developed the most extensive framework for steerability (Teo & Hel-Or, 1998; Hel-Or & Teo, 1996), and proposed the first approach for non-Abelian groups starting with exact steerability

for the largest Abelian subgroup and incrementally steering for the remaining subgroups. Cohen & Welling (2016a); Jacobsen et al. (2017) recently combined steerability and learnable filters.

The most recent "hand-crafted" approach to equivariant representations is the scattering transform (Sifre & Mallat, 2013) which composes rotated and dilated wavelets. Similar to SIFT (Lowe, 2004) this approach relies on the equivariance of anchor points (e.g. the maxima of filtered responses in (translation) space). Translation invariance is obtained through the modulus operation which is computed after each convolution. The final scattering coefficient is invariant to translations and equivariant to local rotations and scalings.

Laptev et al. (2016) achieve transformation invariance by pooling feature maps computed over the input orbit, which scales poorly as it requires forward and backward passes for each orbit element.

Within the context of CNNs, methods of enforcing equivariance fall to two main veins. In the first, equivariance is obtained by constraining filter structure similarly to Lie generator based approaches (Segman et al., 1992; Hel-Or & Teo, 1996). Harmonic Networks (Worrall et al., 2016) use filters derived from the complex harmonics achieving both rotational and translational equivariance. The second requires the use of a filter orbit which is itself equivariant to obtain group equivariance. Cohen & Welling (2016b) convolve with the orbit of a learned filter and prove the equivariance of group-convolutions and preservation of rotational equivariance in the presence of rectification and pooling. Dieleman et al. (2015) process elements of the image orbit individually and use the set of outputs for classification. Gens & Domingos (2014) produce maps of finite-multiparameter groups, Zhou et al. (2017) and Marcos et al. (2016) use a rotational filter orbit to produce oriented feature maps and rotationally invariant features, and Lenc & Vedaldi (2015) propose a transformation layer which acts as a group-convolution by first permuting then transforming by a linear filter.

Our approach, PTN, is akin to the second vein. We achieve global rotational equivariance and expand the notion of CNN equivariance to include scaling. PTN employs log-polar coordinates (canonical coordinates in Segman et al. (1992)) to achieve rotation-dilation group-convolution through translational convolution subject to the assumption of an image center estimated similarly to the STN. Most related to our method is Henriques & Vedaldi (2016), which achieves equivariance by warping the inputs to a fixed grid, with no learned parameters.

When learning features from 3D objects, invariance to transformations is usually achieved through augmenting the training data with transformed versions of the inputs (Wu et al., 2015), or pooling over transformed versions during training and/or test (Maturana & Scherer, 2015; Qi et al., 2016). Sedaghat et al. (2016) show that a multi-task approach, i.e. prediction of both the orientation and class, improves classification performance. In our extension to 3D object classification, we explicitly learn representations equivariant to rotations around a family of parallel axes by transforming the input to cylindrical coordinates about a predicted axis.

## 3 THEORETICAL BACKGROUND

This section is divided into two parts, the first offers a review of equivariance and group-convolutions. The second offers an explicit example of the equivariance of group-convolutions through the 2D similarity transformations group, SIM(2), comprised of translations, dilations and rotations. Reparameterization of SIM(2) to canonical coordinates allows for the application of the SIM(2) group-convolution using translational convolution.

### 3.1 GROUP EQUIVARIANCE

Equivariant representations are highly sought after as they encode both class and deformation information in a predictable way. Let $G$ be a transformation group and $L_g I$ be the group action applied to an image $I$. A mapping $\Phi : E \to F$ is said to be equivariant to the group action $L_g$, $g \in G$ if

$$\Phi(L_g I) = L'_g(\Phi(I)) \tag{1}$$

where $L_g$ and $L'_g$ correspond to application of $g$ to $E$ and $F$ respectively and satisfy $L_{gh} = L_g L_h$. Invariance is the special case of equivariance where $L'_g$ is the identity. In the context of image classification and CNNs, $g \in G$ can be thought of as an image deformation and $\Phi$ a mapping from the image to a feature map.

The inherent translational equivariance of CNNs is independent of the convolutional kernel and evident in the corresponding translation of the output in response to translation of the input. Equivariance to other types of deformations can be achieved through application of the *group-convolution*, a generalization of translational convolution. Letting $f(g)$ and $\phi(g)$ be real valued functions on $G$ with $L_h f(g) = f(h^{-1}g)$, the group-convolution is defined Kyatkin & Chirikjian (2000)

$$(f \star_G \phi)(g) = \int_{h \in G} f(h)\phi(h^{-1}g)\,dh. \tag{2}$$

A slight modification to the definition is necessary in the first CNN layer since the group is acting on the image. The group-convolution reduces to translational convolution when $G$ is translation in $\mathbb{R}^n$ with addition as the group operator,

$$
\begin{aligned}
(f \star \phi)(x) &= \int_h f(h)\phi(h^{-1}x)\,dh \\
&= \int_h f(h)\phi(x - h)\,dh.
\end{aligned}
\tag{3}
$$

Group-convolution requires integrability over a group and identification of the appropriate measure $dg$. It can be proved that given the measure $dg$, group-convolution is always group equivariant:

$$
\begin{aligned}
(L_a f \star_G \phi)(g) &= \int_{h \in G} f(a^{-1}h)\phi(h^{-1}g)\,dh \\
&= \int_{b \in G} f(b)\phi((ab)^{-1}g)\,db \\
&= \int_{b \in G} f(b)\phi(b^{-1}a^{-1}g)\,db \\
&= (f \star_G \phi)(a^{-1}g) \\
&= L_a((f \star_G \phi))(g).
\end{aligned}
\tag{4}
$$

This is depicted in response of an equivariant representation to input deformation (Figure 2 (left)).

## 3.2 Equivariance in $\mathrm{SIM}(2)$

A similarity transformation, $\rho \in \mathrm{SIM}(2)$, acts on a point in $x \in \mathbb{R}^2$ by

$$\rho x \to s\,R\,x + t \quad s \in \mathbb{R}^+,\ R \in SO(2),\ t \in \mathbb{R}^2, \tag{5}$$

where $SO(2)$ is the rotation group. To take advantage of the standard planar convolution in classical CNNs we decompose a $\rho \in \mathrm{SIM}(2)$ into a translation, $t$ in $\mathbb{R}^2$ and a dilated-rotation $r$ in $SO(2) \times \mathbb{R}^+$.

Equivariance to $\mathrm{SIM}(2)$ is achieved by learning the center of the dilated rotation, shifting the original image accordingly then transforming the image to canonical coordinates. In this reparameterization the standard translational convolution is equivalent to the dilated-rotation group-convolution.

The origin predictor is an application of STN to global translation prediction (Jaderberg et al., 2015), the centroid of the output is taken as the origin of the input.

Transformation of the image $L_t I = I(t - t_0)$ (canonization in Soatto (2013)) reduces the $\mathrm{SIM}(2)$ deformation to a dilated-rotation if $t_o$ is the true translation. After centering, we perform $SO(2) \times \mathbb{R}^+$ convolutions on the new image $I_o = I(x - t_o)$:

$$f(r) = \int_{x \in \mathbb{R}^2} I_o(x)\phi(r^{-1}x)\,dx \tag{6}$$

and the feature maps $f$ in subsequent layers

$$h(r) = \int_{s \in SO(2) \times \mathbb{R}^+} f(s)\phi(s^{-1}r)\,ds \tag{7}$$

where $r, s \in SO(2) \times \mathbb{R}^+$. We compute this convolution through use of canonical coordinates for Abelian Lie-groups (Segman et al., 1992). The centered image $I_o(x, y)^1$ is transformed to log-polar coordinates, $I(e^\xi \cos(\theta), e^\xi \sin(\theta))$ hereafter written $\lambda(\xi, \theta)$ with $(\xi, \theta) \in SO(2) \times \mathbb{R}^+$ for

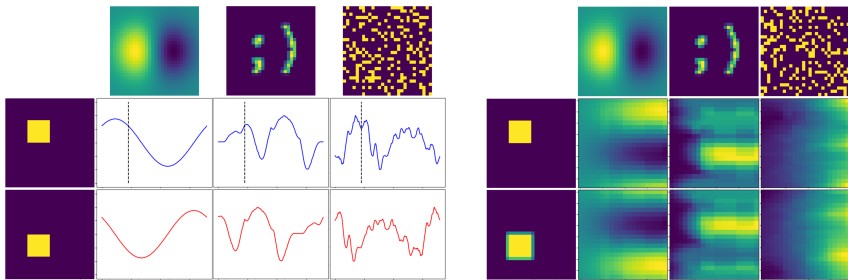

Figure 2: **Left:** Group-convolutions in $SO(2)$. The images in the left most column differ by $90°$ rotation, the filters are shown in the top row. Application of the rotational group-convolution with an arbitrary filter results is shown to produce an equivariant representation. The inner-product each of filter orbit (rotated from $0-360°$) and the image is plotted in blue for the top image and red for the bottom image. Observe how the filter response is shifted by $90°$. **Right:** Group-convolutions in SO(2) $\times \mathbb{R}^+$. Images in the left most column differ by a rotation of $\pi/4$ and scaling of $1.2$. Careful consideration of the resulting heatmaps (shown in canonical coordinates) reveals a shift corresponding to the deformation of the input image.

notational convenience. The shift of the dilated-rotation equivariant representation in response to input deformation is shown in Figure 2 (right) using canonical coordinates.

In canonical coordinates $s^{-1}r = \xi_r - \xi, \theta_r - \theta$ and the SO(2) $\times \mathbb{R}^+$ group-convolution[2] can be expressed and efficiently implemented as a planar convolution

$$\int_s f(s)\phi(s^{-1}r) \ ds = \int_s \lambda(\xi, \theta)\phi(\xi_r - \xi, \theta_r - \theta) \ d\xi d\theta. \tag{8}$$

To summarize, we (1) construct a network of translational convolutions, (2) take the centroid of the last layer, (3) shift the original image to accordingly, (4) convert to log-polar coordinates, and (5) apply a second network[3] of translational convolutions. The result is a feature map equivariant to dilated-rotations around the origin.

## 4  ARCHITECTURE

PTN is comprised of two main components connected by the polar transformer module. The first part is the polar origin predictor and the second is the classifier (a conventional fully convolutional network). The building block of the network is a $3 \times 3 \times K$ convolutional layer followed by batch normalization, an ReLU and occasional subsampling through strided convolution. We will refer to this building block simply as *block*. Figure 3 shows the architecture.

### 4.1  POLAR ORIGIN PREDICTOR

The polar origin predictor operates on the original image and comprises a sequence of blocks followed by a $1 \times 1$ convolution. The output is a single channel feature map, the centroid of which is taken as the origin of the polar transform.

There are some difficulties in training a neural network to predict coordinates in images. Some approaches (Toshev & Szegedy, 2014) attempt to use fully connected layers to directly regress the coordinates with limited success. A better option is to predict heatmaps (Tompson et al., 2014; Newell et al., 2016), and take their argmax. However, this can be problematic since backpropagation gradients are zero in all but one point, which impedes learning.

---

[1]we abuse the notation here and momentarily we use $x$ as the x-coordinate instead of $x \in \mathbb{R}^2$.

[2]abuse of the term, SO(2) $\times \mathbb{R}^+$ is not a group because the dilation $\xi$ is not compact.

[3]the network employs rectifier and pooling which have been shown to preserve equivariance (Cohen & Welling, 2016b).

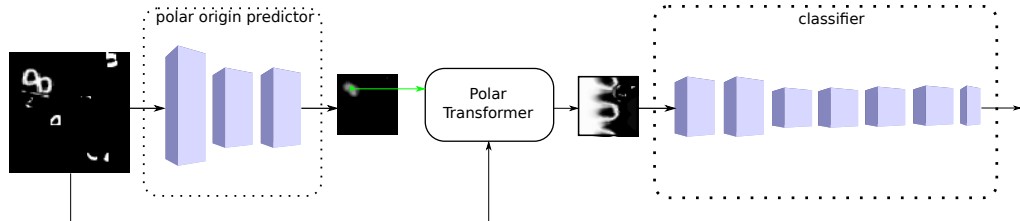

Figure 3: Network architecture. The input image passes through a fully convolutional network, the polar origin predictor, which outputs a heatmap. The centroid of the heatmap (two coordinates), together with the input image, goes into the polar transformer module, which performs a polar transform with origin at the input coordinates. The obtained polar representation is invariant with respect to the original object location; and rotations and dilations are now shifts, which are handled equivariantly by a conventional classifier CNN.

The usual approach to heatmap prediction is evaluation of a loss against some ground truth. In this approach the argmax gradient problem is circumvented by supervision. In PTN the the gradient of the output coordinates must be taken with respect to the heatmap since the polar origin is unknown and must be learned. Use of argmax is avoided by using the centroid of the heatmap as the polar origin. The gradient of the centroid with respect to the heatmap is constant and nonzero for all points, making learning possible.

## 4.2 Polar transformer module

The polar transformer module takes the origin prediction and image as inputs and outputs the log-polar representation of the input. The module uses the same differentiable image sampling technique as STN (Jaderberg et al., 2015), which allows output coordinates $V_i$ to be expressed in terms of the input $U$ and the source sample point coordinates $(x_i^s, y_i^s)$. The log-polar transform in terms of the source sample points and target regular grid $(x_i^t, y_i^t)$ is:

$$x_i^s = x_0 + r^{x_i^t/W} \cos \frac{2\pi y_i^t}{H} \tag{9}$$

$$y_i^s = y_0 + r^{x_i^t/W} \sin \frac{2\pi y_i^t}{H} \tag{10}$$

where $(x_0, y_0)$ is the origin, $W, H$ are the output width and height, and $r$ is the maximum distance from the origin, set to $0.5\sqrt{H^2 + W^2}$ in our experiments.

## 4.3 Wrap-around padding

To maintain feature map resolution, most CNN implementations use zero-padding. This is not ideal for the polar representation, as it is periodic about the angular axis. A rotation of the input result in a vertical shift of the output, wrapping at the boundary; hence, identification of the top and bottom most rows is most appropriate. This is achieved with wrap-around padding on the vertical dimension. The top most row of the feature map is padded using the bottom rows and vice versa. Zero-padding is used in the horizontal dimension. Table 5 shows a performance evaluation.

## 4.4 Polar origin augmentation

To improve robustness of our method, we augment the polar origin during training time by adding a random shift to the regressed polar origin coordinates. Note that this comes for little computational cost compared to conventional augmentation methods such as rotating the input image. Table 5 quantifies the performance gains of this kind of augmentation.

# 5 EXPERIMENTS

## 5.1 ARCHITECTURES

We briefly define the architectures in this section, see A for details. CCNN is a conventional fully convolutional network; PCNN is the same, but applied to polar images with central origin. STN is our implementation of the spatial transformer networks (Jaderberg et al., 2015). PTN is our polar transformer networks, and PTN-CNN is a combination of PTN and CCNN. The suffixes S and B indicate small and big networks, according to the number of parameters. The suffixes + and ++ indicate training and training+test rotation augmentation.

We perform rotation augmentation for polar-based methods. In theory, the effect of input rotation is just a shift in the corresponding polar image, which should not affect the classifier CNN. In practice, interpolation and angle discretization effects result in slightly different polar images for rotated inputs, so even the polar-based methods benefit from this kind of augmentation.

## 5.2 ROTATED MNIST (LAROCHELLE ET AL., 2007)

Table 1 shows the results. We divide the analysis in two parts; on the left, we show approaches with smaller networks and no rotation augmentation, on the right there are no restrictions.

Between the restricted approaches, the Harmonic Network (Worrall et al., 2016) outperforms the PTN by a small margin, but with almost 4x more training time, because the convolutions on complex variables are more costly. Also worth mentioning is the poor performance of the STN with no augmentation, which shows that learning the transformation parameters is much harder than learning the polar origin coordinates.

Between the unrestricted approaches, most variants of PTN-B outperform the current state of the art, with significant improvements when combined with CCNN and/or test time augmentation.

Finally, we note that the PCNN achieves a relatively high accuracy in this dataset because the digits are mostly centered, so using the polar transform origin as the image center is reasonable. Our method, however, outperforms it by a high margin, showing that even in this case, it is possible to find an origin away from the image center that results in a more distinctive representation.

Table 1: Performance on rotated MNIST. Errors are averages of several runs, with standard deviations within parenthesis. Times are average training time per epoch.

| Model | error [%] | params | time [s] | Model | error [%] | params | time [s] |
|---|---|---|---|---|---|---|---|
| PTN-S | 1.83 (0.04) | 27k | 3.64 (0.04) | PTN-B+ | 1.14 (0.08) | 129k | 4.38 (0.02) |
| PCNN-S | 2.6 (0.08) | 22k | 2.61 (0.04) | PTN-B++ | 0.95 (0.09) | 129k | 4.38[6] |
| CCNN-S | 5.76 (0.35) | 22k | 2.43 (0.02) | PTN-CNN-B+ | 1.01 (0.06) | 254k | 7.36 |
| STN-S | 7.87 (0.18) | 43k | 3.90 (0.05) | PTN-CNN-B++ | **0.89 (0.06)** | 254k | 7.36[6] |
| HNet [1] | **1.69** | 33k | 13.29 (0.19) | PCNN-B+ | 1.37 (0.00) | 124k | 3.30 (0.04) |
| P4CNN [2] | 2.28 | 22k | - | CCNN-B+ | 1.53 (0.07) | 124k | 2.98 (0.02) |
| | | | | STN-B+ | 1.31 (0.05) | 146k | 4.57 (0.04) |
| | | | | OR-TIPooling [3] | 1.54 | $\approx$ 1M | - |
| | | | | TI-Pooling [4] | 1.2 | $\approx$ 1M | 42.90 |
| | | | | RotEqNet [5] | 1.01 | 100k | - |

[1, 2, 3, 4, 5] Worrall et al. (2016); Cohen & Welling (2016b); Zhou et al. (2017); Laptev et al. (2016); Marcos et al. (2016)

[6] Test time performance is 8x slower when using test time augmentation

## 5.3 OTHER MNIST VARIANTS

We also perform experiments in other MNIST variants. MNIST R, RTS are replicated from Jaderberg et al. (2015). We introduce SIM2MNIST, with a more challenging set of transformations from SIM(2). See B for more details about the datasets.

Table 2 shows the results. We can see that the PTN performance mostly matches the STN on both MNIST R and RTS. The deformations on these datasets are mild and data is plenty, so the performance may be saturated.

On SIM2MNIST, however, the deformations are more challenging and the training set 5x smaller. The PCNN performance is significantly lower, which reiterates the importance of predicting the best

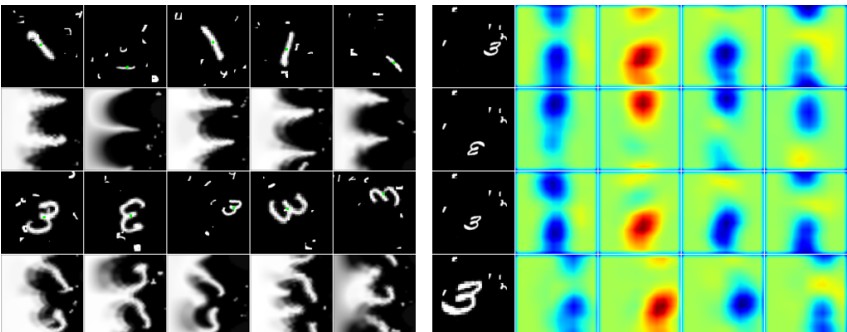

Figure 4: **Left:** The rows alternate between samples from SIM2MNIST, where the predicted origin is shown in green, and their learned polar representation. Note how rotations and dilations of the object become shifts. **Right:** Each row shows a different input and correspondent feature maps on the last convolutional layer. The first and second rows show that the $180°$ rotation results in a half-height vertical shift of the feature maps. The third and fourth rows show that the $2.4\times$ dilation results in a shift right of the feature maps. The first and third rows show invariance to translation.

polar origin. The HNet outperforms the other methods (except the PTN), thanks to its translation and rotation equivariance properties. Our method is more efficient both in number of parameters and training time, and is also equivariant to dilations, achieving the best performance by a large margin.

Table 2: Performance on MNIST variants.

| | MNIST R | | | MNIST RTS | | | SIM2MNIST[1] | | |
| | error [%] | pars | time | error [%] | pars | time | error [%] | pars | time |
| --- | --- | --- | --- | --- | --- | --- | --- | --- | --- |
| PTN-S+ | 0.88 (0.04) | 29k | 19.72 | 0.78 (0.05) | 32k | 24.48 | 5.44 (0.03) | 35k | 11.92 |
| PTN-B+ | 0.62 (0.04) | 129k | 20.37 | 0.57 (0.03) | 134k | 28.74 | **5.03 (0.11)** | 134k | 12.02 |
| PCNN-B+ | 0.81 (0.04) | 124k | 13.97 | 0.70 (0.01) | 129k | 17.19 | 15.46 (0.22) | 129k | 5.33 |
| CCNN-B+ | 0.74 (0.01) | 124k | 12.79 | 0.62 (0.07) | 129k | 15.97 | 11.73 (0.57) | 129k | 5.28 |
| STN-B+ | **0.61 (0.02)** | 146k | 23.12 | 0.54 (0.02) | 150k | 27.90 | 12.35 (1.61) | 150k | 10.41 |
| STN (Jaderberg et al., 2015) | 0.7 | 400k | - | **0.5** | 400k | - | - | - | - |
| HNet [2](Worrall et al., 2016) | - | - | - | - | - | - | 9.28 (0.05) | 44k | 31.42 |
| TI-Pooling (Laptev et al., 2016) | 0.8 | $\approx$ 1M | - | - | - | - | - | - | - |

[1] No augmentation is used with SIM2MNIST, despite the + suffixes

[2] Our modified version, with two extra layers with subsampling to account for larger input

## 5.4 VISUALIZATION

We visualize network activations to confirm our claims about invariance to translation and equivariance to rotations and dilations.

Figure 4 (left) shows some of the predicted polar origins and the results of the polar transform. We can see that the network learns to reject clutter and to find a suitable origin for the polar transform, and that the representation after the polar transformer module does present the properties claimed.

We proceed to visualize if the properties are preserved in deeper layers. Figure 4 (right) shows the activations of selected channels from the last convolutional layer, for different rotations, dilations, and translations of the input. The reader can verify that the equivariance to rotations and dilations, and the invariance to translations are indeed preserved during the sequence of convolutional layers.

## 5.5 EXTENSION TO 3D OBJECT CLASSIFICATION

We extend our model to perform 3D object classification from voxel occupancy grids. We assume that the inputs are transformed by random rotations around an axis from a family of parallel axes. Then, a rotation around that axis corresponds to a translation in cylindrical coordinates.

In order to achieve equivariance to rotations, we predict an axis and use it as the origin to transform to cylindrical coordinates. If the axis is parallel to one of the input grid axes, the cylindrical transform

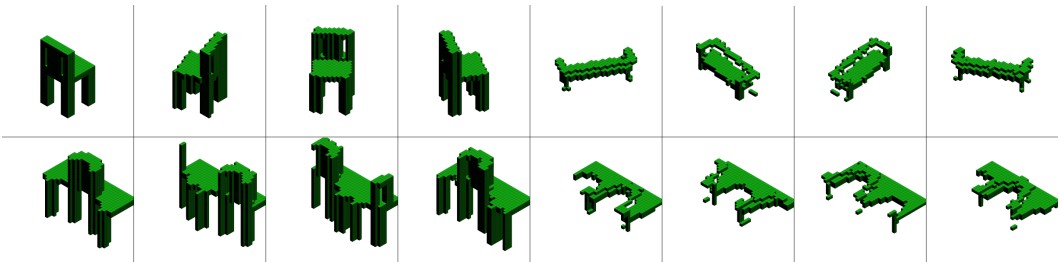

Figure 5: Top: rotated voxel occupancy grids. Bottom: corresponding cylindrical representations. Note how rotations around a vertical axis correspond to translations over a horizontal axis.

amounts to channel-wise polar transforms, where the origin is the same for all channels and each channel is a 2D slice of the 3D voxel grid. In this setting, we can just apply the polar transformer layer to each slice.

We use a technique similar to the anisotropic probing of Qi et al. (2016) to predict the axis. Let $z$ denote the input grid axis parallel to the rotation axis. We treat the dimension indexed by $z$ as channels, and run regular 2D convolutional layers, reducing the number of channels on each layer, eventually collapsing to a single 2D heatmap. The heatmap centroid gives one point of the axis, and the direction is parallel to $z$. In other words, the centroid is the origin of all channel-wise polar transforms. We then proceed with a regular 3D CNN classifier, acting on the cylindrical representation. The 3D convolutions are equivariant to translations; since they act on cylindrical coordinates, the learned representation is equivariant to input rotations around axes parallel to $z$.

We run experiments on ModelNet40 (Wu et al., 2015), which contains objects rotated around the gravity direction ($z$). Figure 5 shows examples of input voxel grids and their cylindrical coordinates representation, while table 3 shows the classification performance. To the best of our knowledge, our method outperforms all published voxel-based methods, even with no test time augmentation. However, the multi-view based methods generally outperform the voxel-based. (Qi et al., 2016).

Note that we could also achieve equivariance to scale by using log-cylindrical or log-spherical coordinates, but none of these change of coordinates would result in equivariance to arbitrary 3D rotations.

Table 3: ModelNet40 classification performance. We compare only with voxel-based methods.

| Model | Avg. class accuracy [%] | Avg. instance accuracy [%] |
| --- | --- | --- |
| Cylindrical Transformer (Ours) | **86.5** | 89.9 |
| 3D ShapeNets (Wu et al., 2015) | 77.3 | - |
| VoxNet (Maturana & Scherer, 2015) | 83 | - |
| MO-SubvolumeSup (Qi et al., 2016) | 86.0 | 89.2 |
| MO-Aniprobing (Qi et al., 2016) | 85.6 | 89.9 |

## 6 CONCLUSION

We have proposed a novel network whose output is invariant to translations and equivariant to the group of dilations/rotations. We have combined the idea of learning the translation (similar to the spatial transformer) but providing equivariance for the scaling and rotation, avoiding, thus, fully connected layers required for the pose regression in the spatial transformer. Equivariance with respect to dilated rotations is achieved by convolution in this group. Such a convolution would require the production of multiple group copies, however, we avoid this by transforming into canonical coordinates. We improve the state of the art performance on rotated MNIST by a large margin, and outperform all other tested methods on a new dataset we call SIM2MNIST. We expect our approach to be applicable to other problems, where the presence of different orientations and scales hinder the performance of conventional CNNs.

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

APPENDICES

## A   ARCHITECTURES DETAILS

We implement the following architectures for comparison,

- **Conventional CNN (CCNN)**, a fully convolutional network, composed of a sequence of convolutional layers and some rounds of subsampling .
- **Polar CNN (PCNN)**, same architecture as CCNN, operating on polar images. The log-polar transform is pre-computed at the image center before training, as in Henriques & Vedaldi (2016). The fundamental difference between our method and this is that we learn the polar origin implicitly, instead of fixing it.
- **Spatial Transformer Network (STN)**, our implementation of Jaderberg et al. (2015), replacing the localization network by four blocks of 20 filters and stride 2, followed by a 20 unit fully connected layer, which we found to perform better. The transformation regressed is in SIM(2), and a CCNN comes after the transform.
- **Polar Transformer Network (PTN)**, our proposed method. The polar origin predictor comprises three blocks of 20 filters each, with stride 2 on the first block (or the first two blocks, when input is $96 \times 96$). The classification network is the CCNN.
- **PTN-CNN**, we classify based on the sum of the per class scores of instances of PTN and CCNN trained independently.

The following suffixes qualify the architectures described above:

- **S**, "small" network, with seven blocks of 20 filters and one round of subsampling (equivalent to the Z2CNN in Cohen & Welling (2016b)).
- **B**, "big" network, with 8 blocks with the following number of filters: 16, 16, 32, 32, 32, 64, 64, 64. Subsampling by strided convolution is used whenever the number of filters increase. We add up to two 2 extra blocks of 16 filters with stride 2 at the beginning to handle larger input resolutions (one for $42 \times 42$ and two for $96 \times 96$).
- **+**, training time rotation augmentation by continuous angles.
- **++**, training and test time rotation augmentation. We input 8 rotated versions the the query image and classify using the sum of the per class scores.

**Cylindrical transformer network:** The axis prediction part of the cylindrical transformer network is composed of four 2D blocks, with $5 \times 5$ kernels and 32, 16, 8, and 4 channels, no subsampling. The classifier is composed of eight 3D convolutional blocks, with $3 \times 3 \times 3$ kernels, the following number of filters: 32, 32, 32, 64, 64, 64, 128, 128, and subsampling whenever the number of filters increase. Total number of params is approximately 1M.

## B   DATASET DETAILS

- **Rotated MNIST** The rotated MNIST dataset (Larochelle et al., 2007) is composed of $28 \times 28$, $360°$ rotated images of handwritten digits. The training, validation and test sets are of sizes 10k, 2k, and 50k, respectively.
- **MNIST R**, we replicate it from Jaderberg et al. (2015). It has 60k training and 10k testing samples, where the digits of the original MNIST are rotated between $[-90°, 90°]$. It is also know as half-rotated MNIST (Laptev et al., 2016).
- **MNIST RTS**, we replicate it from Jaderberg et al. (2015). It has 60k training and 10k testing samples, where the digits of the original MNIST are rotated between $[-45°, 45°]$, scaled between 0.7 and 1.2, and shifted within a $42 \times 42$ black canvas.
- **SIM2MNIST**, we introduce a more challenging dataset, based on MNIST, perturbed by random transformations from SIM(2). The images are $96 \times 96$, with $360°$ rotations; the scale factors range from 1 to 2.4, and the digits can appear anywhere in the image. The training, validation and test set have size 10k, 5k, and 50k, respectively.

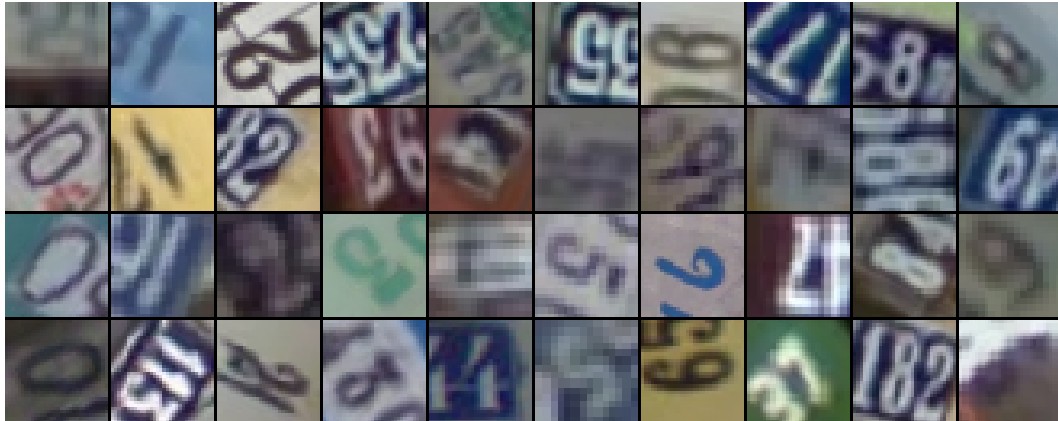

Figure 6: ROTSVHN samples. Since the digits are cropped from larger images, no artifacts are introduced when rotating. The 6s and 9s are indistinguishable when rotated. Note that there are usually visible digits on the sides, which pose a challenge for classification and PTN origin prediction.

Table 4: SVHN classification performance. The minus suffix indicate removal of 6s and 9s. PTN shows slightly worse performance on the unperturbed dataset, but is clearly superior when rotations are present.

|  | SVHN | ROTSVHN | SVHN- | ROTSVHN- |
|---|---|---|---|---|
| PTN-ResNet32 (Ours) | 2.82 (0.07) | **7.90 (0.14)** | 2.85 (0.07) | **3.96 (0.04)** |
| ResNet32 | **2.25 (0.15)** | 9.83 (0.29) | **2.09 (0.06)** | 5.39 (0.09) |

## C  SVHN Experiments

In order to demonstrate the efficacy of PTN on real-world RGB images, we run experiments on the Street View House Numbers (SVHN) dataset Netzer et al. (2011), and a rotated version that we introduce (ROTSVHN) . The dataset contains cropped images of single digits, as well as the slightly larger images from where the digits are cropped. Using the latter, we can extract the rotated digits without introducing artifacts. Figure 6 shows some examples from the ROTSVHN.

We use a 32 layer Residual Network (He et al., 2016) as a baseline (ResNet32). The PTN-ResNet32 has 8 residual convolutional layers as the origin predictor, followed by a ResNet32.

In contrast with handwritten digits, the 6s and 9s in house numbers are usually indistinguishable. To remove this effect from our analysis, we also run experiments removing those classes from the datasets (which is denoted by appending a minus to the dataset name). Table 4 shows the results.

The reader will note that rotations cause a significant performance loss on the conventional ResNet; the error increases from 2.09% to 5.39%, even when removing 6s and 9s from the dataset. With PTN, on the other hand, the error goes from 2.85% to 3.96%, which shows our method is more robust to the perturbations, although the performance on the unperturbed datasets is slightly worse. We expect the PTN to be even more advantageous when large scale variations are also present.

## D  Ablation Study

We quantify the performance boost obtained with wrap around padding, polar origin augmentation, and training time rotation augmentation. Results are based on the PTN-B variant trained on Rotated MNIST. We remove one operation at a time and verify that the performance consistently drops, which indicates that all operations are indeed helpful. Table 5 shows the results.

Table 5: Ablation study. Rotation and polar origin augmentation during training time, and wrap around padding all contribute to reduce the error. Results are from PTN-B on the rotated MNIST.

| Origin aug. | Rotation aug. | Wrap padding | Error [%] |
|---|---|---|---|
| Yes | Yes | Yes | 1.12 (0.03) |
| No | Yes | Yes | 1.33 (0.12) |
| Yes | No | Yes | 1.46 (0.11) |
| Yes | Yes | No | 1.31 (0.06) |

