# OpenReview forum: "Polar Transformer Networks"
_ICLR.cc/2018/Conference — Accept (Poster)_

### Official Review · AnonReviewer2 · 2017-11-26
**Interesting approach to equivariant CNNs**

**Rating:** 7
**Confidence:** 4

**Review:**

This paper proposes a method to learn networks invariant to translation and equivariant to rotation and scale of arbitrary precision. The idea is to jointly train
- a network predicting a polar origin,
- a module transforming the image into a log-polar representation according to the predicted origin,
- a final classifier performing the desired classification task.
A (not too large) translation of the input image therefore does not change the log-polar representation.
Rotation and scale from the polar origin result in translation of the log-polar representation. As convolutions are translation equivariant, the final classifier becomes rotation and scale equivariant in terms of the input image. Rotation and scale can have arbitrary precision, which is novel to the best of my knowledge.

(+) In my opinion, this is a simple, attractive approach to rotation and scale equivariant CNNs.

(-) The evaluation, however, is quite limited. The approach is evaluated on:
 1) several variants of MNIST. The authors introduce a new variant (SIM2MNIST), which is created by applying random similitudes to the images from MNIST. This variant is of course very well suited to the proposed method, and a bit artificial.
 2) 3d voxel occupancy grids with a small resolution. The objects can be rotated around the z-axis, and the method is used to be equivariant to this rotation.

(-) Since the method starts by predicting the polar origin, wouldn't it be possible to also predict rotation and scale? Then the input image could be rectified to a canonical orientation and scale, without needing equivariance. My intuition is that this simpler approach would work better. It should at least be evaluated.

Despite these weaknesses, I think this paper should be interesting for researchers looking into equivariant CNNs.

---

> ### Author Response · Authors · 2017-12-17
> **Response to AnonReviewer2**
>
> Thank you for the review.
>
> > (-) The evaluation, however, is quite limited.
>
> Please check Appendix C for the newly included results on the Street
> View House Numbers dataset (SVHN), which shows that our method is also
> applicable to real-world RGB images.  We show superior performance
> than the baselines when perturbations are present.
>
> > (-) Since the method starts by predicting the polar
> > origin, wouldn't it be possible to also predict rotation
> > and scale? Then the input image could be rectified to a
> > canonical orientation and scale, without needing
> > equivariance. My intuition is that this simpler approach
> > would work better. It should at least be evaluated.
>
> Your suggestion seems to be what is done in the Spatial Transformer
> Networks (STN) (Jaderberg et al).  Our experiments show that
> regressing scale and rotation angle is a hard problem, requiring more
> data and larger networks; on the other hand, learning the coordinates
> of a single point as a heatmap centroid is easier.  We show direct
> comparison of our method and the STN on tables 1 and 2. The advantage
> of our method is significant, specially with small number of samples
> (rotated MNIST, SIM2MNIST) and large perturbations (SIM2MNIST).

---

### Official Review · AnonReviewer3 · 2017-11-26
**Good idea but not there yet**

**Rating:** 7
**Confidence:** 4

**Review:**

This paper presents a new convolutional network architecture that is invariant to global translations and equivariant to rotations and scaling. The method is combination of a spatial transformer module that predicts a focal point, around which a log-polar transform is performed. The resulting log-polar image is analyzed by a conventional CNN.

I find the basic idea quite compelling. Although this is not mentioned in the article, the proposed approach is quite similar to human vision in that people choose where to focus their eyes, and have an approximately log-polar sampling grid in the retina. Furthermore, dealing well with variations in scale is a long-standing and difficult problem in computer vision, and using a log-spaced sampling grid seems like a sensible approach to deal with it.

One fundamental limitation of the proposed approach is that although it is invariant to global translations, it does not have the built-in equivariance to local translations that a ConvNet has. Although we do not have data on this, I would guess that for more complex datasets like imagenet / ms coco, where a lot of variation can be reasonably well modelled by diffeomorphisms, this will result in degraded performance.

The use of the heatmap centroid as the prediction for the focal point is potentially problematic as well. It would not work if the heatmap is multimodal, e.g. when there are multiple instances in the same image or when there is a lot of clutter.

There is a minor conceptual confusion on page 4, where it is written that "Group-convolution requires integrability over a group and identification of the appropriate measure dg. We ignore this detail as implementation requires application of the sum instead of integral."
When approximating an integral by a sum, one should generally use quadrature weights that depend on the measure, so the measure cannot be ignored. Fortunately, in the chosen parameterization, the Haar measure is equal to the standard Lebesque measure, and so when using equally-spaced sampling points in this parameterization, the quadrature weights should be one. (Please double-check this - I'm only expressing my mathematical intuition but have not actually proven this).

It does not make sense to say that "The above convolution requires computation of the orbit which is feasible with respect to the finite rotation group, but not for general rotation-dilations", and then proceed to do exactly that (in canonical coordinates). Since the rotation-dilation group is 2D, just like the 2D translation group used in ConvNets, this is entirely feasible. The use of canonical coordinates is certainly a sensible choice (for the reason given above), but it does not make an infeasible computation feasible.

The authors may want to consider citing
- Warped Convolutions: Efficient Invariance to Spatial Transformations, Henriques & Vedaldi.
This paper also uses a log-polar transform, but lacks the focal point prediction / STN.
Likewise, although the paper makes a good effort to rewiev the literature on equivariance / steerability, it missed several recent works in this area:
- Steerable CNNs, Cohen & Welling
- Dynamic Steerable Blocks in Deep Residual Networks, Jacobsen et al.
- Learning Steerable Filters for Rotation Equivariant CNNs, Weiler et al.
The last paper reports 0.71% error on MNIST-rot, which is slightly better than the PTN-CNN-B++ reported on in this paper.

The experimental results presented in this paper are quite good, but both MNIST and ModelNet40 seem like simple / toyish datasets. For reasons outlined above, I am not convinced that this approach in its current form would work very well on more complicated problems. If the authors can show that it does (either in its current form or after improving it, e.g. with multiple saccades, or other improvements) I would recommend this paper for publication.


Minor issues & typos
- Section 3.1, psi_gh = psi_g psi_h. I suppose you use psi for L and L', but this is not very clear.
- L_h f = f(h^{-1}), p. 4
- "coordiantes", p. 5

---

> ### Author Response · Authors · 2017-12-17
> **Response to AnonReviewer3**
>
> Thank you for the review and insightful comments.
>
> > One fundamental limitation of the proposed approach is
> > that although it is invariant to global translations, it
> > does not have the built-in equivariance to local
> > translations that a ConvNet has. Although we do not have
> > data on this, I would guess that for more complex datasets
> > like imagenet / ms coco, where a lot of variation can be
> > reasonably well modelled by diffeomorphisms, this will
> > result in degraded performance.
>
> We trade-off local translation equivariance for global roto-dilation
> equivariance.  It is likely that for imagenet/coco local translation
> equivariance is more important, but that may not be the case when
> global rotations and large scale variance is present.  We show that
> the trade-off favors roto-dilation equivariance for the rotated MNIST,
> ModelNet40, and rotated SVHN (included in the appendix during the
> review period); a more interesting real life example where
> roto-dilation equivariance could be beneficial is object recognition
> in satellite images.
>
> > The use of the heatmap centroid as the prediction for the
> > focal point is potentially problematic as well. It would
> > not work if the heatmap is multimodal, e.g. when there are
> > multiple instances in the same image or when there is a
> > lot of clutter.
>
> Our method assumes that there is a single correct instance per image,
> so the multiple instance case could indeed be a problem.  Note that in
> the newly included SVHN results there are often multiple instances
> present (see figure 6); however, the correct label is the one of the
> central digit and our origin predictor learns to use that.  If we
> assume that multiple instances may be present in different positions,
> with no central prior, we need to treat the problem as multiple object
> detection.  Since the origin predictor is fully convolutional, we
> believe we could train our model on single objects and test it with
> multiple objects, with some sort of test-time non-maximum-suppression
> on the final heatmap.  It is likely that a soft argmax (perhaps with a
> loss term enforcing concentration) should be used instead of computing
> the centroid.  We could also pre-train the origin predictor with
> object center supervision, and then fine-tune it end-to-end (since we
> have shown that the object center is not necessarily the best origin).
> Note that we have not tried this approach yet, though we did
> experiment with the soft argmax and concentration loss in the
> single-instance setting, and found no performance difference.
>
> > There is a minor conceptual confusion on page 4, where it
> > is written that "Group-convolution requires integrability
> > over a group and identification of the appropriate measure
> > dg. We ignore this detail as implementation requires
> > application of the sum instead of integral." When
> > approximating an integral by a sum, one should generally
> > use quadrature weights that depend on the measure, so the
> > measure cannot be ignored.
>
> This sentence is incorrect and was removed: "We ignore this detail as
> implementation requires application of the sum instead of integral.",
> thanks for bringing it to our attention.  Equation (8) is determined
> to be true by applying the definition of the Haar measure for the
> dilated-rotation group and a change of coordinates to log-polar,
> showing that the quadrature weights are indeed one.  We have updated
> the text accordingly.
>
>
> > It does not make sense to say that "The above convolution
> > requires computation of the orbit which is feasible with
> > respect to the finite rotation group, but not for general
> > rotation-dilations", and then proceed to do exactly that
> > (in canonical coordinates). Since the rotation-dilation
> > group is 2D, just like the 2D translation group used in
> > ConvNets, this is entirely feasible. The use of canonical
> > coordinates is certainly a sensible choice (for the reason
> > given above), but it does not make an infeasible
> > computation feasible.
>
> True, implying that the computation is 'infeasible' is wrong. We have
> updated the text.
>
> > The authors may want to consider citing (..)
>
> Thanks for pointing these out, we have updated our citations.  Note
> that Weiler et al. only appeared on 11/21, almost a month after the
> submission deadline, and that we did cite Henriques & Vedaldi,
> although mistakenly not in "Related Work" (already fixed).
>
> > For reasons outlined above, I am not convinced that this
> > approach in its current form would work very well on more
> > complicated problems. If the authors can show that it does
> > (either in its current form or after improving it,
> > e.g. with multiple saccades, or other improvements) I
> > would recommend this paper for publication.
>
> Please check Appendix C for the newly included results on the Street
> View House Numbers dataset (SVHN), which shows that our method is also
> applicable to real-world RGB images.  We show superior performance
> than the baselines when perturbations are present.

---

> > ### Comment · AnonReviewer3 · 2018-01-12
> > **Revised**
> >
> > I think my initial review score was a bit low. There is certainly still a lot of residual uncertainty about whether the method in its current state would work well on more serious vision problems, but:
> > 1) The method is conceptually novel and innovative
> > 2) I can see a plausible path towards real-world usage. This may require some further ideas for how to deal with multiple objects, how to learn where to focus, etc., but this paper doesn't have to solve everything at once.
> > So I recommend the paper for acceptance.

---

### Official Review · AnonReviewer1 · 2017-11-27
**Very nice and potentially influential paper whose practical usefulness remains to be shown**

**Rating:** 8
**Confidence:** 3

**Review:**

The authors introduce the Polar Transformer, a special case of the Spatial Transformer (Jaderberg et al. 2015) that achieves rotation and scale equivariance by using a log-polar sampling grid. The paper is very well written, easy to follow and substantiates its claims convincingly on variants of MNIST. A weakness of the paper is that it does not attempt to solve a real-world problem. However, I think because it is a conceptually novel and potentially very influential idea, it is a valuable contribution as it stands.

Issues:

- The clutter in SIM2MNIST is so small that predicting the polar origin is essentially trivially solved by a low-pass filter. Although this criticism also applies to most previous work using ‘cluttered’ variants of MNIST, I still think it needs to be considered. What happens if predicting the polar origin is not trivial and prone to errors? These presumably lead to catastrophic failure of the post-transformer network, which is likely to be a problem in any real-world scenario.

- I’m not sure if Section 5.5 strengthens the paper. Unlike the rest of the paper, it feels very ‘quick & dirty’ and not very principled. It doesn’t live up to the promise of rotation and scale equivariance in 3D. If I understand it correctly, it’s simply a polar transformer in (x,y) with z maintained as a linear axis and assumed to be parallel to the axis of rotation. This means that the promise of rotation and scale equivariance holds up only along (x,y). I guess it’s not possible to build full 3D rotation/scale equivariance with the authors’ approach (spherical coordinates probably don’t do the job), but at least the scale equivariance could presumably have been achieved by using log-spaced samples along z and predicting the origin in 3D. So instead of showing a quick ‘hack’, I would have preferred an honest discussion of the limitations and maybe a sketch of a path forward even if no implemented solution is provided.

---

> ### Author Response · Authors · 2017-12-17
> **Response to AnonReviewer1**
>
> Thank you for the review.
>
> > A weakness of the paper is that it does not attempt to
> > solve a real-world problem.
>
> Please check Appendix C for the newly included results on the Street
> View House Numbers dataset (SVHN), which shows that our method is also
> applicable to real-world RGB images.  We show superior performance
> than the baselines when perturbations are present.
>
> > The clutter in SIM2MNIST is so small that predicting the
> > polar origin is essentially trivially solved by a low-pass
> > filter. Although this criticism also applies to most
> > previous work using ‘cluttered’ variants of MNIST, I still
> > think it needs to be considered. What happens if
> > predicting the polar origin is not trivial and prone to
> > errors? These presumably lead to catastrophic failure of
> > the post-transformer network, which is likely to be a
> > problem in any real-world scenario.
>
> While we agree that the amount of clutter could be overcome by
> hand-designed methods, we argue that learning the origin in an
> end-to-end fashion is advantageous since, in this case, the origin is
> learned precisely for classification of the log-polar representation.
> This is quantified in Table 1, which compares our method (PTN), with
> the Polar CNN (PCNN), which fixes the origin at the image center.  The
> results show that even though the digits on the rotated MNIST are
> centered, the learned origin results in significant improvements.
>
> In more challenging scenarios, it is likely that a deeper origin
> predictor network or more sophisticated object detection model would
> be necessary.  For example, we could pre-train the origin predictor
> with object center supervision, and then fine-tune it end-to-end.  For
> the newly included SVHN experiments we used a deeper residual origin
> predictor, but no pre-training was necessary.
>
> > I’m not sure if Section 5.5 strengthens the paper. Unlike
> > the rest of the paper, it feels very ‘quick & dirty’ and
> > not very principled. It doesn’t live up to the promise of
> > rotation and scale equivariance in 3D. If I understand it
> > correctly, it’s simply a polar transformer in (x,y) with z
> > maintained as a linear axis and assumed to be parallel to
> > the axis of rotation. This means that the promise of
> > rotation and scale equivariance holds up only along
> > (x,y). I guess it’s not possible to build full 3D
> > rotation/scale equivariance with the authors’ approach
> > (spherical coordinates probably don’t do the job), but at
> > least the scale equivariance could presumably have been
> > achieved by using log-spaced samples along z and
> > predicting the origin in 3D. So instead of showing a quick
> > ‘hack’, I would have preferred an honest discussion of the
> > limitations and maybe a sketch of a path forward even if
> > no implemented solution is provided.
>
> Your understanding is correct.  The purpose of section 5.5 is to show
> that our method is applicable to a more challenging problem, from a
> completely different domain.  Even though our implementation may be
> considered a hack (applying channel-wise polar transforms), the
> concept of using cylindrical coordinates for azimuthal rotation
> equivariance is solid, and so is learning axis of the transform.  This
> is the direct extension of the PTN to 3D.
>
> As you mentioned, it is not possible to achieve full SO(3)
> equivariance using cylindrical or spherical coordinates.  Hence, we
> aim for equivariance to rotations around axes parallel to z.  We
> consider this a reasonable assumption, which is equivalent to assuming
> sensors parallel to the ground, or known gravity direction in robotics
> applications, for example.  Moreover, the vast majority of results on
> ModelNet40 are with azimuthal rotation only.
>
> It is indeed the case that scale equivariance could be achieved by
> log-spaced samples along z.  It could also be achieved by using
> spherical coordinates.  We experimented with those but neither improve
> performance on ModelNet40, since the scale variability is negligible
> on it.  We included these considerations in the text of section 5.5.

---

### Author Response · Authors · 2018-01-05
**List of changes in revised version**

- Included Appendix C, with experiments on the Street View House Numbers dataset (SVHN),
- fixed/clarified math issues raised by AnonReviewer3,
- included citations suggested by AnonReviewer3,
- included clarification in section 5.5, to address issues raised by AnonReviewer1, and
- rearranged paragraphs and removed redundant sentences to maintain number of pages.

---

### Public Comment · (anonymous) · 2018-02-26
**Ignores Vast majority of voxel-based ModelNet40 Benchmarks**

Hi all,

No issues with the paper otherwise, but just wanted to point out that this completely ignores the publicly available ModelNet leaderboards (http://modelnet.cs.princeton.edu), which show a variety of voxel-based classifiers that outperform the results presented here.

---

> ### Author Response · Authors · 2018-02-26
> **Please be more specific**
>
> Thanks for your interest. Could you please be a bit more specific? I revisited the ModelNet leaderboard and it seems that only two volumetric methods currently outperforms ours:
> 1) LightNet, which we were not aware of (it is listed as accepted on 10/23, 4 days before ICLR deadline),
> 2) VRN, which is an ensemble, hence an unfair comparison.
>
> Please let me know if I'm missing something. Note that there is usually a lot of confusion when comparing ModelNet results, since many papers make different assumptions (i.e., accuracy per instance or per class, or taking subsets of the dataset).

---

### Decision · Program_Chairs · 2018-01-29
**ICLR 2018 Conference Acceptance Decision**

**Decision:**

Accept (Poster)

**Comment:**

The paper proposes a new deep architecture based on polar transformation for improving rotational invariance. The proposed method is interesting and the experimental results strong classification performance on small/medium-scale datasets (e.g., rotated MNIST and its variants with added translations and clutters, ModelNet40, etc.). It will be more impressive and impactful if the proposed method can bring performance improvement on large-scale, real datasets with potentially cluttered scenes (e.g., Imagenet, Pascal VOC, MS-COCO, etc.).